# Effect of physiotherapy interventions on pain management, function and quality of life in patellofemoral pain syndrome: A systematic review protocol

Chaitali S. Vikhe[1]*, Swapnil U. Ramteke[1], Sharath Hullumani[2]

**1** Department of Sports Physiotherapy, Ravi Nair Physiotherapy College, Datta Meghe Institute Higher Education and Research (DU), Sawangi (Meghe), Wardha, India, **2** Assistant Professor Department of Paediatric Physiotherapy, Ravi Nair Physiotherapy College, Datta Meghe Institute of Higher Education and Research (DU) Sawangi Meghe, Maharashtra, Wardha, India

* chaitalivikhe619@gmail.com

**Data Availability Statement:** No datasets were generated or analysed during the current study.

**Funding:** The author(s) received no specific funding for this work.

## Abstract

### Background

Patellofemoral pain syndrome (PFPS) is a prevalent and often debilitating condition, affecting approximately 22.7% of the population and significantly contributing to knee-related disorders. It primarily impacts young athletes and sedentary individuals, impairing their quality of life and functional abilities. Despite extensive research, the optimal management strategies for PFPS remain contentious due to its multifactorial etiology and persistent symptoms. The findings of this review will guide clinical practice and future research, ultimately improving outcomes for individuals with PFPS.

### Objective

This systematic review aims to evaluate the effectiveness of physiotherapy interventions in alleviating pain, enhancing functional outcomes, and improving the quality of life among runners who have been diagnosed with PFPS.

### Methods

This review will be carried out on randomized controlled trials (rcts) that assess any type of physiotherapy intervention for PFPS in runners. Eligible studies must report on pain intensity, functional improvement, or quality of life outcomes. Searches will be conducted in pubmed/MEDLINE, Cochrane Library, and pedro, covering publications from May 2014 to April 2024. Two reviewers will independently screen and select studies, extract data, and assess quality using the Cochrane Risk of Bias tool. We will perform a narrative synthesis of the data, focusing on pain management, functional recovery, and quality of life improvements. Sensitivity and subgroup analyses will explore variations across study designs and participant characteristics.

**Competing interests:** The authors have declared that no competing interests exist.

## Conclusion

This systematic review protocol seeks to identify effective physiotherapy interventions for managing PFPS in runners. By analyzing rcts, the review will provide evidence-based recommendations to reduce pain, improve function, and enhance quality of life. The findings will guide clinical decisions

## Systematic review registration

International Prospective Register of Systematic Reviews (PROSPERO) under ID: CRD42024531888.

Date- 7/05/2024, URL **https://www.crd.york.ac.uk/prospero/#recorddetails**.

## Introduction

Patellofemoral pain syndrome (PFPS) is a prevalent condition, affecting 22.7% of the population and accounting for 25–40% of all knee disorders [1, 2]. It is characterized by pain around the patella during activities such as squatting, running, and climbing steps, impacting both athletes and sedentary individuals [3]. It is one of the most common causes of knee pain, impacting both athletes and sedentary individuals [4] PFPS represents 11–17% of knee pain cases in general practice and a significant 25–40% in sports injury clinics, underscoring its substantial impact across different medical settings [5] Managing PFPS can be challenging, with symptoms persisting in approximately 50% of individuals and sometimes lasting for decades [6], The multifactorial nature of PFPS complicates identifying the exact cause of pain in each individual [7].

While regular physical activity has health benefits, it also carries the risk of musculoskeletal injuries [8] PFPS significantly impacts the quality of life, sometimes comparably to osteoarthritis, particularly concerning for young people. Several factors are linked to PFPS, including a larger quadriceps angle, dynamic knee valgus, and increased rear-foot eversion during heel strike [9, 10]. Running, though beneficial, poses injury PFPS represents 11–17% of knee pain cases in general practice and a significant 25–40% in sports injury clinics, underscoring its substantial impact across different medical settings risks, with occurrence rates between 19% and 78%. PFPS is prevalent among long-distance runners, often limiting performance and return to sport [11, 12].

Despite extensive research, there is no consensus on the exact etiology of PFPS or the most effective treatment strategies [13]. This lack of agreement has led to PFPS being described as a "black hole" in orthopedic medicine, due to the absence of a single therapeutic approach that comprehensively addresses all aspects of patellofemoral dysfunction. Approximately 70–90% of individuals with PFPS experience recurrent or chronic symptoms, underscoring the need for effective management strategies [14, 15].

Given the high incidence of running-related musculoskeletal injuries due to overuse [16], there is a critical need for a systematic review to evaluate the effectiveness of these interventions. Previous systematic reviews have examined various aspects of PFPS management; however, a focused review specifically addressing physiotherapy interventions for runners is lacking. This systematic review aims to identify and assess the most effective physiotherapy interventions for managing PFPS in runners, to prevent recurrence and facilitate a successful return to sports. By consolidating evidence from randomized controlled trials (RCTs), this review will provide valuable insights and evidence-based recommendations for clinicians, as well as identify areas requiring further research.

### Review question

What are the most effective physiotherapy interventions for managing patellofemoral pain syndrome (PFPS) in runners, and how do these interventions compare in terms of reducing pain, improving function, and preventing recurrence?

## Methods

### Eligibility criteria

The eligibility criteria for this systematic review are based on the PICO principle and include:

1. **Population:** Studies must involve runners diagnosed with patellofemoral pain syndrome (PFPS). There are no age restrictions, and both male and female participants from any geographical location are considered.

2. **Intervention:** The review will include studies evaluating a range of interventions for PFPS, such as:

   ○ Exercise therapy (e.g., strengthening, stretching)

   ○ Manual therapy (e.g., joint mobilizations, soft tissue techniques)

   ○ Biomechanical interventions (e.g., orthotics, gait retraining)

   ○ Electrotherapy modalities (e.g., ultrasound, TENS)

3. **Comparison:** Studies must have a comparator group, which could be:

   • Placebo

   • Standard care

   • No intervention

   • Other active treatments

4. **Outcome Measures:** Studies should report on at least one of the following outcomes:

   • Pain intensity (using scales such as the Visual Analog Scale or Numeric Rating Scale)

   • Functional improvement (measured with tools like the Knee Injury and Osteoarthritis Outcome Score or the Lower Extremity Functional Scale)

   • Quality of life (assessed with standardized questionnaires or scales)

5. **Study Design:** The focus will be on randomized controlled trials (rcts).

6. **Language:** Only studies published in English will be included.

7. **Publication Date:** Studies published from May 2014 to April 2024 will be considered.

8. **Publication Type:** Both published studies and gray literature (e.g., conference abstracts, theses) will be included if relevant.

### Information sources

Databases to be searched include pubmed/MEDLINE, Cochrane Library, and Pedro, covering articles published in English from May 2014 to April 2024. Google Scholar will be used to supplement the search but is recognized as a search engine rather than a formal database.

## Search strategy

A comprehensive search will be conducted in pubmed/MEDLINE, Cochrane Library, Pedro, and Google Scholar. Medical Subject Headings (mesh) terms and relevant keywords, such as "patellofemoral pain syndrome," "runners," "interventions," and "physiotherapy," will be utilized. Searches will be re-run before the final analysis to ensure the inclusion of the most recent studies. Efforts will also be made to locate unpublished studies through trial registers, investigator contacts, and conference proceedings.

## Selection process

A flowchart depicting the study selection process will be included. Two reviewers will independently screen titles and abstracts for relevance. Full-text articles of studies deemed potentially relevant will be reviewed for eligibility. Discrepancies regarding eligibility will be resolved through discussion or by consulting a third reviewer.

## Data collection process

Two reviewers will independently extract data using a standardized data extraction form, which will be pilot-tested before full data extraction. This form will capture study characteristics, participant demographics, intervention and comparator details, outcome measures, and results. Any discrepancies or uncertainties will be resolved through discussion or by consulting a third reviewer if necessary.

## Data items

Participant characteristics will include age (mean age and age range), gender distribution, inclusion and exclusion criteria, and baseline characteristics such as health status, duration of PFPS, level of physical activity, and other relevant demographic or clinical information. Intervention details will encompass the type of physiotherapy intervention (e.g., exercise therapy, manual therapy, biomechanical interventions, electrotherapy modalities, patient education), the duration of the intervention, its frequency and intensity, and the delivery method (e.g., individual sessions, group sessions, home-based programs). For comparators, data on the type of control or comparator interventions (e.g., placebo, standard care, no intervention), their duration, frequency, and intensity will be collected. Outcome measures will focus on pain intensity (using scales such as the Visual Analog Scale or Numeric Rating Scale), functional improvement (assessed with tools like the Knee Injury and Osteoarthritis Outcome Score and the Lower Extremity Functional Scale), quality of life (evaluated using standardized questionnaires or scales), and any adverse effects reported. Study characteristics will include the study design (e.g., randomized controlled trial, cohort study), sample size, funding sources to assess potential bias, and any reported conflicts of interest by the study authors. Pre-planned data assumptions and simplifications will be applied where necessary to ensure consistent and accurate data extraction and analysis across all included studies.

## Outcomes

Pain intensity will be assessed by continuous scales, such as the Visual Analog Scale (VAS) or the Numeric Pain Rating Scale (NPRS), with changes from baseline to post-intervention and follow-up time points analyzed. Functional improvements, including enhancements in knee function, range of motion, strength, and performance related to running, will be evaluated by validated measures, such as the Knee Injury and Osteoarthritis Outcome Score (KOOS) and the Lower Extremity Functional Scale (LEFS). Quality of life will be evaluated using

standardized questionnaires or scales, focusing on overall well-being and satisfaction with life post-intervention.

### Risk of bias assessment

The methodological quality and risk of bias in included studies will be evaluated using the Cochrane Risk of Bias tool for rcts. For studies with different designs, such as observational studies, the ROBINS-I tool (Risk Of Bias In Non-randomized Studies—of Interventions) will be used to assess bias. To address publication bias, a funnel plot will be used if there are sufficient studies (usually more than 10), and Egger's test or Begg's test will be conducted for statistical analysis of asymmetry in the funnel plot.

### Data synthesis

To evaluate primary outcomes such as pain, quality of life, and functional improvement, standardized mean differences (SMD) or relative risks (rrs) with 95% confidence intervals (cis) will be used to measure treatment effectiveness for PFPS. Given the anticipated heterogeneity among studies, data synthesis will primarily involve a narrative approach. Findings from included studies will be summarized descriptively, highlighting key outcomes, intervention types, and their reported effects on pain management, functional recovery, and quality of life. A narrative summary will be complemented by graphical representations to enhance clarity. While a meta-analysis is not planned due to heterogeneity, sensitivity analyses will explore the impact of study design, quality, and risk of bias on reported outcomes. Subgroup analyses will be conducted to assess differences in intervention effects across participant demographics or intervention characteristics.

## Confidence in cumulative evidence

This systematic review will be evaluated using the GRADE (Grading of Recommendations, Assessment, Development, and Evaluation) approach. The GRADE approach will have the potential to conduct a rigorous assessment of the cumulative evidence related to the effectiveness of physiotherapy interventions for managing Patellofemoral Pain Syndrome in runners.

## Discussion

This protocol outlines a comprehensive approach to evaluating physiotherapy interventions for the management of patellofemoral pain syndrome (PFPS) in runners. PFPS, a prevalent condition among athletes, significantly impairs the quality of life due to its multifactorial origin and high recurrence rates. Despite its prevalence, effective management remains elusive, largely due to the lack of consensus on the ideal treatment approach. Physiotherapy, with its non-invasive and holistic strategies, targets underlying biomechanical abnormalities, muscular imbalances, and pain, offering a promising avenue for treatment. The findings from this systematic review are expected to provide evidence-based recommendations that will influence clinical practice by guiding treatment plans to optimize patient outcomes. Additionally, this review may inform evidence-based guidelines and identify areas for future research, advancing our understanding and management of PFPS.

### 1. Contribution to the Field

This systematic review is anticipated to make a significant contribution to the field by synthesizing current evidence on the effectiveness of various physiotherapy interventions for managing PFPS in runners. By systematically analyzing the outcomes of different interventions, this

review will equip clinicians with an evidence-based understanding of which treatments are most effective in reducing pain, improving function, and enhancing the quality of life for runners suffering from PFPS.

The findings will not only inform clinical practice but will also identify gaps in the current research, guiding future studies. Specifically, this review aims to clarify the relative effectiveness of different interventions, which can lead to more tailored treatment strategies and potentially improved outcomes. Furthermore, this review's insights may contribute to the development of standardized treatment protocols and evidence-based clinical guidelines, thus enhancing the overall quality of care for individuals with PFPS.

## 2. Anticipated challenges

One of the primary challenges anticipated in this review is the heterogeneity of the included studies. Variability in study designs, types of interventions, outcome measures, and participant characteristics may limit the ability to conduct a meta-analysis and synthesize the data quantitatively. This heterogeneity could complicate the interpretation of findings and reduce the ability to draw definitive conclusions about the effectiveness of specific interventions.

To address this challenge, the review will employ a comprehensive narrative synthesis approach, allowing for a more nuanced discussion of the results in light of these variations. Where feasible, subgroup and sensitivity analyses will be conducted to explore the impact of specific variables, such as study design, intervention type, and participant demographics, on the outcomes. These analyses will help to elucidate the conditions under which certain interventions are most effective.

Another anticipated challenge is the potential for publication bias, which could influence the review's findings. To mitigate this, we will include extensive search strategies that encompass gray literature and unpublished studies, ensuring a broader and more representative data set. Additionally, statistical methods such as funnel plots and Egger's test will be utilized to assess and account for publication bias where appropriate.

By proactively addressing these challenges through robust methodological strategies, this systematic review aims to provide a thorough and reliable synthesis of the available evidence, thereby contributing valuable insights into the effective management of PFPS in runners.

## Conclusion

This systematic review protocol aims to identify effective physiotherapy interventions for managing PFPS in runners. By analysing rcts, the review will offer evidence-based recommendations to reduce pain, improve function, and enhance quality of life. The findings will guide clinical practice and future research, ultimately improving outcomes for individuals with PFPS.

## Supporting information

**S1 Checklist. PRISMA-P (Preferred Reporting Items for Systematic review and Meta-Analysis Protocols) 2015 checklist: Recommended items to address in a systematic review protocol*.**
(DOCX)

**S1 Data.**
(CSV)

## Acknowledgments

This section briefly acknowledges the invaluable assistance of specific colleagues, institutions, or agencies that supported the author's efforts.

## Author Contributions

**Conceptualization:** Chaitali S. Vikhe, Swapnil U. Ramteke, Sharath Hullumani.

**Data curation:** Chaitali S. Vikhe.

**Funding acquisition:** Sharath Hullumani.

**Investigation:** Chaitali S. Vikhe, Swapnil U. Ramteke, Sharath Hullumani.

**Methodology:** Chaitali S. Vikhe.

**Project administration:** Chaitali S. Vikhe.

**Software:** Sharath Hullumani.

**Supervision:** Swapnil U. Ramteke, Sharath Hullumani.

**Validation:** Chaitali S. Vikhe, Swapnil U. Ramteke, Sharath Hullumani.

**Visualization:** Chaitali S. Vikhe, Swapnil U. Ramteke, Sharath Hullumani.

**Writing – original draft:** Chaitali S. Vikhe.

**Writing – review & editing:** Chaitali S. Vikhe.

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
