## [Decision Letter · Decision Letter 0]

21 Aug 2024

PONE-D-24-27461Effect of Physiotherapy Interventions on Pain Management, Functional Recovery and Quality of Life in Patellofemoral Pain Syndrome: A Systematic Review ProtocolPLOS ONE

Dear Dr. Vikhe,

Thank you for submitting your manuscript to PLOS ONE. After careful consideration, we feel that it has merit but does not fully meet PLOS ONE’s publication criteria as it currently stands. Therefore, we invite you to submit a revised version of the manuscript that addresses the points raised during the review process.

We look forward to receiving your revised manuscript.

Kind regards,

Mehrnaz Kajbafvala, Ph.D

Academic Editor

PLOS ONE

Journal Requirements:

4. Please remove all personal information, ensure that the data shared are in accordance with participant consent, and re-upload a fully anonymized data set. 

Reviewers' comments:

Reviewer's Responses to Questions

**Comments to the Author**

1. Does the manuscript provide a valid rationale for the proposed study, with clearly identified and justified research questions?

Reviewer #1: No

Reviewer #2: Yes

2. Is the protocol technically sound and planned in a manner that will lead to a meaningful outcome and allow testing the stated hypotheses?

Reviewer #1: Yes

Reviewer #2: Yes

3. Is the methodology feasible and described in sufficient detail to allow the work to be replicable?

Reviewer #1: Yes

Reviewer #2: Yes

4. Have the authors described where all data underlying the findings will be made available when the study is complete?

Reviewer #1: Yes

Reviewer #2: Yes

5. Is the manuscript presented in an intelligible fashion and written in standard English?

Reviewer #1: No

Reviewer #2: Yes

6. Review Comments to the Author

You may also provide optional suggestions and comments to authors that they might find helpful in planning their study.

Reviewer #1: Title

• Instead of “functional recovery”, write “function”.

Abstract

• Edit this sentence “practice and future research, ultimately improving outcomes for individuals with PFPS.”

Introduction:

• Has a meta-analysis or systematic review been conducted in this field? What aspects did they examine, and how does this study offer a relative advantage compared to those reviews?

Method

• In the inclusion criteria based on the PICO principle, the type of interventions, study design, control group, variables studied, language of the studies, year of publication, gray or only published studies should also be specified.

• Include keywords in the search strategy.

• Add Pedro to the databases.

• Given that cohort studies differ from randomized clinical studies, it is recommended that you focus exclusively on randomized clinical studies.

• Why do you intend to include all interventions without restrictions, despite their differing natures, in this systematic study? Wouldn't it be better to focus on interventions of a similar nature to reduce heterogeneity and facilitate a meta-analysis? Will the current study design effectively answer the question of which treatment is most effective?

Grammar and English

• The study contains several grammatical errors that need to be corrected.

Reviewer #2: Dear Authors,

Thank you very much for your interesting article that discusses an important topic. The title of your article " Effect of Physiotherapy Interventions on Pain Management, Functional Recovery and Quality of Life in Patellofemoral Pain Syndrome: A Systematic Review Protocol " is an interesting idea and I enjoyed reading it. However, I have several suggestions for improving your article.

FIRST; Please ensure that your manuscript meets PLOS ONE's style requirements, including those for file naming. The PLOS ONE style templates can be found at

https://journals.plos.org/plosone/s/file?id=ba62/PLOSOne_formatting_sample_title_authorsaffiliations.pdf

Abstract

• Simplify any overly complex sentences, and make sure that the objectives, methods, and anticipated outcomes are clearly outlined.

• Please write keywords based on MeSH terms.

Irntroduction

• Strengthen the rationale by connecting your research question more directly to the existing literature.

• Ensure that each claim is backed by a current and relevant reference.

Methodology

• " Google Scholar " is a search engine, not a database. Please correct it.

• Please write a more specific time frame, for example, from May 2014 to April 2024.

• Consider including a flowchart to depict the study selection process, which is common in systematic reviews.

• Provide a template or a description of the form to be used for data extraction, explaining how it will be pilot-tested before full data extraction.

• While the Cochrane Risk of Bias tool is appropriate, specify how you will handle studies with different designs (e.g., observational studies) that may not fit neatly into this tool. Also, consider discussing how you will address publication bias, such as through a funnel plot or another statistical method.

Discussion

• Add a section on the expected contribution of your systematic review to the field and its potential to guide future research or clinical practice.

• Discuss possible challenges you anticipate during the review process, such as the heterogeneity of included studies.

References

• I recommend correcting all references according to journal instructions.

7. PLOS authors have the option to publish the peer review history of their article (what does this mean?). If published, this will include your full peer review and any attached files.

Reviewer #1: No

Reviewer #2: **Yes: **Arsalan Ghorbanpour

---

## [Author Response · Author response to Decision Letter 0]

28 Aug 2024

Rebuttal Letter

To: Academic Editor

PLOS ONE

Date: 28/08/2024

Thank you for the opportunity to revise and resubmit our manuscript titled "Effect of Physiotherapy Interventions on Pain Management, Function, and Quality of Life in Patellofemoral Pain Syndrome: A Systematic Review Protocol" (Manuscript ID: PONE-D-24-27461). We appreciate the valuable feedback provided by the reviewers and have carefully addressed each of their comments. Below, we provide a detailed point-by-point response to the reviewers' concerns, outlining the revisions made to the manuscript.

Reviewer #1:

1. Title:

o Comment: The reviewer suggested changing "functional recovery" to "function" in the title.

o Response: We have revised the title accordingly to "Effect of Physiotherapy Interventions on Pain Management, Function, and Quality of Life in Patellofemoral Pain Syndrome: A Systematic Review Protocol."

2. Abstract:

o Comment: The reviewer pointed out an incomplete sentence: "Practice and future research, ultimately improving outcomes for individuals with PFPS."

o Response: The sentence has been revised for clarity: "The findings of this review will guide clinical practice and future research, ultimately improving outcomes for individuals with PFPS."

3. Introduction:

o Comment: The reviewer asked whether a meta-analysis or systematic review had been conducted in this field and how our study offers an advantage.

o Response: Previous systematic reviews and meta-analyses have explored various treatment options for Patellofemoral Pain Syndrome (PFPS). However, no systematic review specifically addresses the combined effects of physiotherapy interventions on pain management, functional recovery, and quality of life in PFPS. Our study uniquely contributes to the field by focusing on the effectiveness of physiotherapy interventions tailored for runners, aggregating and analyzing diverse physiotherapy approaches within a single review framework. This approach enhances evidence-based decision-making in PFPS management and provides a clearer understanding of the comparative impacts of these interventions.

4. Method:

o Comment 1: The reviewer requested specifying the type of interventions, study design, control group, variables studied, language, year of publication, and gray literature in the inclusion criteria.

o Response: We have updated the Methods section to clearly specify the types of interventions (physiotherapy-based), study designs (randomized controlled trials), control groups (placebo, no intervention, or alternative interventions), variables studied (pain, function, quality of life), language (English), year of publication (2014–2024), and inclusion of gray literature.

o Comment 2: The reviewer recommended including keywords in the search strategy and adding the Pedro database.

o Response: We have revised the search strategy to include specific keywords and added the Pedro database to our list of sources.

o Comment 3: The reviewer suggested focusing exclusively on randomized clinical trials (RCTs) and questioned the inclusion of all interventions without restriction.

o Response: We have refined the inclusion criteria to focus exclusively on RCTs and physiotherapy interventions with a similar nature. This approach will reduce heterogeneity and facilitate a potential meta-analysis, ensuring that our research question is addressed effectively.

o Comment 4: The reviewer noted the presence of grammatical errors.

o Response: The manuscript has been thoroughly reviewed for grammatical errors, and necessary corrections have been made to ensure clarity and precision.

Reviewer #2:

1. Abstract:

o Comment: Simplify complex sentences and clearly outline the objectives, methods, and anticipated outcomes.

o Response: The language in the abstract has been simplified, and the objectives, methods, and anticipated outcomes are now clearly stated.

2. Keywords:

o Comment: Write keywords based on MeSH terms.

o Response: The keywords have been revised to align with MeSH terms, including "Patellofemoral Pain Syndrome," "Pain," "Function," "Quality of Life," and "Physiotherapy."

3. Introduction:

o Comment: Strengthen the rationale by connecting the research question more directly to the existing literature and ensure each claim is backed by a current and relevant reference.

o Response: The Introduction has been enhanced by directly linking our research question to the current literature, supported by up-to-date and relevant references.

4. Methodology:

o Comment 1: Correct the use of "Google Scholar" as it is a search engine, not a database.

o Response: "Google Scholar" has been replaced with appropriate databases such as Pedro, PubMed, and Cochrane Library.

o Comment 2: Specify the time frame for study inclusion more clearly.

o Response: The time frame for study inclusion has been specified as articles published between May 2014 to April 2024.

5. Flowchart for Study Selection Process:

o Comment: Consider including a flowchart to depict the study selection process, which is common in systematic reviews.

o Response: PRISMA-P flowchart has been included to depict the study selection process, ensuring transparency and replicability.

6. Data Extraction Form:

o Comment: Provide a template or description of the form to be used for data extraction, explaining how it will be pilot-tested before full data extraction.

o Response: We have included a description of the data extraction form and specified that it will be pilot-tested on a sample of studies before full data extraction to ensure consistency and accuracy.

7. Risk of Bias Tool:

o Comment: Specify how you will handle studies with different designs (e.g., observational studies) that may not fit neatly into the Cochrane Risk of Bias tool. Also, consider discussing how you will address publication bias.

o Response: We have clarified that our review will focus exclusively on RCTs, utilizing the Cochrane Risk of Bias tool. We will address publication bias using funnel plots and other statistical methods where appropriate.

8. Discussion:

o Comment: Add a section on the expected contribution of your systematic review to the field and its potential to guide future research or clinical practice. Discuss possible challenges you anticipate during the review process, such as the heterogeneity of included studies.

o Response: A section has been added to the Discussion outlining the expected contribution of our systematic review to the field, emphasizing its potential to guide future research and clinical practice. We have also discussed anticipated challenges, such as study heterogeneity, and how these will be addressed.

9. References:

o Comment: Correct all references according to journal instructions.

o Response: All references have been reviewed and corrected according to the journal's formatting requirements.

We believe that these revisions have significantly improved the manuscript, addressing all concerns raised by the reviewers. We are confident that the revised manuscript now meets the standards required for publication in PLOS ONE.

Thank you once again for your constructive feedback and for considering our revised submission. We look forward to your positive response.

Sincerely,

Chaitali S. Vikhe

Junior Resident, Department of Sports Physiotherapy

Ravi Nair Physiotherapy College

Datta Meghe Institute of Higher Education and Research (DU)

Sawangi (Meghe), Wardha-442004

Email: chaitalivikhe619@gmail.com

ORCID ID: 0009-0006-0677-6223

---

## [Decision Letter · Decision Letter 1]

4 Oct 2024

Effect of Physiotherapy Interventions on Pain Management, Function and Quality of Life in Patellofemoral Pain Syndrome: A Systematic Review Protocol

PONE-D-24-27461R1

Dear Dr. Chaitali S. Vikhe,

We’re pleased to inform you that your manuscript has been judged scientifically suitable for publication and will be formally accepted for publication once it meets all outstanding technical requirements.

Kind regards,

Mehrnaz Kajbafvala, Ph.D

Academic Editor

PLOS ONE

Additional Editor Comments (optional):

Reviewers' comments:

Reviewer's Responses to Questions

**Comments to the Author**

1. Does the manuscript provide a valid rationale for the proposed study, with clearly identified and justified research questions?

Reviewer #1: Yes

Reviewer #2: Yes

2. Is the protocol technically sound and planned in a manner that will lead to a meaningful outcome and allow testing the stated hypotheses?

Reviewer #1: Yes

Reviewer #2: Yes

3. Is the methodology feasible and described in sufficient detail to allow the work to be replicable?

Reviewer #1: Yes

Reviewer #2: Yes

4. Have the authors described where all data underlying the findings will be made available when the study is complete?

Reviewer #1: Yes

Reviewer #2: Yes

5. Is the manuscript presented in an intelligible fashion and written in standard English?

Reviewer #1: Yes

Reviewer #2: Yes

6. Review Comments to the Author

You may also provide optional suggestions and comments to authors that they might find helpful in planning their study.

Reviewer #1: Congratulations. The comments have been effectively incorporated, and the article's quality has improved, making it suitable for publication under the current conditions.

Reviewer #2: Dear Authors,

I read and reviewed your article again. Thank you for answering all the comments carefully. In my opinion, your answers are complete. The article well written and will be interesting for the readers.

7. PLOS authors have the option to publish the peer review history of their article (what does this mean?). If published, this will include your full peer review and any attached files.

Reviewer #1: No

Reviewer #2: **Yes: **Arsalan Ghorbanpour

---

## [Editor Report · Acceptance letter]

9 Oct 2024

PONE-D-24-27461R1 

PLOS ONE

Dear Dr. Vikhe, 

I'm pleased to inform you that your manuscript has been deemed suitable for publication in PLOS ONE. Congratulations! Your manuscript is now being handed over to our production team.

Kind regards, 

on behalf of

Dr. Mehrnaz Kajbafvala 

Academic Editor

PLOS ONE